# Factors Determining Effective Probiotic Activity: Evaluation of Survival and Antibacterial Activity of Selected Probiotic Products Using an “In Vitro” Study

**DOI:** 10.3390/nu14163323

**Published:** 2022-08-13

**Authors:** Malgorzata Bernatek, Wioletta Żukiewicz-Sobczak, Sabina Lachowicz-Wiśniewska, Jacek Piątek

**Affiliations:** 1Faculty of Health Sciences, Calisia University, 62-800 Kalisz, Poland; 2Internal Medicine Department of Hospital in Jarocin, 63-200 Jarocin, Poland

**Keywords:** probiotics, hydrochloric acid, bile, pathogen inhibition zones

## Abstract

There are many different probiotic products on the market. Are they all equally effective? What criteria should a probiotic formulation meet to provide the most benefit to the patient? The current research aims to evaluate the parameters that influence the effectiveness of market probiotic products. These properties are critical for restoring eubiosis in patients with drug-induced dysbiosis or other pathological conditions, which could be caused by stress, wrong eating. Methods: The disintegration time of probiotic capsules in hydrochloric acid was investigated using a disintegration testing device. The survival rate of probiotic preparations in hydrochloric acid at pH 2 and in a 0.4% bile solution was then evaluated. For this purpose, the number of bacteria before and after incubation in the respective solutions was determined using the plate method. Inhibition of gastrointestinal pathogens by the probiotic products was determined using the Strus bar graph method. The highest survival rate of probiotic bacteria at low pH is shown by preparations produced in the form of acid-resistant capsules. Conclusions: The most important factor determining the good survival of bacterial strains under conditions simulating the gastrointestinal tract is the type of capsule used for their production and storage. The best antimicrobial activity against most common human gastrointestinal pathogens such as *Eschericha coli, Shigella, Salmonella* spp., *Clostridioides difficile* (the largest inhibition zones) are shown by probiotic products with the greatest diversity of bacterial strains.

## 1. Introduction

According to the FAO/WHO definition, probiotics are live microorganisms that, when incorporated into the body in adequate amounts, result in health benefits to the host [1,2]. Unfortunately, there are still disturbing reports that some probiotic preparations placed on the market are not fully characterized or tested [3]. Despite the growing knowledge of probiotics, studies do not always pay enough attention to quality aspects. In addition, the choice of probiotics is very large. Few studies have been motivated in their search for probiotics by analyzing the function and action of a particular strain of bacteria, the number of bacterial strains in a capsule or the pharmaceutical technology used by the manufacturer. Additionally, these are the essential factors determining the optimal performance of a given probiotic market product [4]. The environmental conditions in the various parts of the gastrointestinal tract limit the number of live bacteria reaching their destination [5]. The second important factor determining the optimal survival of probiotic bacteria is the method of their production, which determines the number of live bacteria reaching the intestine, resulting in its colonization [6,7]. The bacteria’s passage through the acidic environment of the stomach, which is extremely negative for living bacteria, is crucial for the proper colonization of the intestine by probiotics. After the oral intake of a probiotic product, it reaches the stomach, where it is exposed to hydrochloric acid contained in gastric juice. In the case of probiotics, the effect of hydrochloric acid on the microorganisms present is unfavorable, decreasing their survival [8,9]. In order to protect the bacterial strains in probiotic preparations, various technologies have been introduced. These technologies basically work to protect the bacterial strains from the destructive effects of hydrochloric acid and other digestive factors. Probiotic forms encapsulated in oil or acid-resistant capsules that are supposed to protect *Lactobacilli* [10] or *Bifidobacterium* from the negative effects of enzymes and low pH, are appearing with increasing frequency [9,11].

An important property of the intestinal microbiota, especially for administered probiotic microorganisms, is their resistance to further gastrointestinal conditions, especially their tolerance and growth in the presence of bile salts. Bacteria can utilize several mechanisms of defense against bile, including special transport mechanisms, the synthesis of various surface proteins and fatty acids, or the production of exopolysaccharides [12]. The ability to enzymatically hydrolyze bile salts is found in many bacteria. Cholylglycine hydrolase hydrolyzes bile salts and is a constitutive intracellular enzyme responsible for the hydrolysis of the amide bond between glycine or taurine and the steroid nucleus of bile acids [13]. Its presence has been demonstrated in specific microorganisms from several bacterial types (*Lactobacillus* spp., *Bifidobacterium* spp., *Clostridium* spp., and *Bacteroides* spp.).

Bile salt hydrolase activity may facilitate bile detoxification, provide opportunities for bacteria to utilize released amino acids as a carbon and nitrogen source, or promote cholesterol incorporation into the cell wall. The deconjugation of bile salts may be directly related to the reduction of serum cholesterol, from which conjugated bile salts are synthesized de novo [14]. The ability of microorganisms to assimilate or bind ingested cholesterol to the cell wall or to eliminate it by co-precipitation with released cholic acid has also been documented. Some gut microbiota produce cholesterol reductase, which catalyzes the conversion of cholesterol to insoluble coprostanol that is then excreted in the feces, also reducing exogenous cholesterol [15,16]. As we can see, there is a wide spectrum of defense mechanisms in probiotic microorganisms against bile. It seems obvious that probiotic preparations on the market should be tested for resistance to hydrochloric acid (in an imitation of gastric conditions) and bile, because only after passage through the stomach and duodenum do they reach their proper site of action. 

Another very important factor in choosing the right probiotic is the inhibitory effect on pathological bacteria. One simple test to determine the predicted action is to examine the inhibition zones of probiotics against pathological bacteria [17,18]. At the same time, it should be noted that we can also be faced with dysbiosis when using drugs such as metformin, proton pump inhibitors (PPI), or non-steroidal anti-inflammatory drugs or antibiotics. The medicines mentioned above may increase the risk of developing intestinal pathogens. For example, treatment with PPI could induce *Salmonella* and *C. dificille* infections, while metformin may cause excessive growth of *Escherichia* and *Shigella* spp. Another widely used class of medicines – NSAIDs, can induce dysbiosis and damage the intestinal mucosa by activating inappropriate mechanisms of the non-specific immune response. It should be emphasized that NSAIDs are often used with PPIs, and such a combination may potentiate dysbiosis and increase the risk of *C. dificille* infection [19,20].

The human digestive tract under physiological conditions is colonized by more than 400 different species of bacteria. Their total mass reaches up to 2 kg. It is therefore no surprise that the intestinal microbiota has a huge impact on our health and its disorders can have serious health consequences. Bacterial flora disturbances may occur as a result of infection with pathological bacteria such as *Salmonella, Shigella*, or *Escherichia coli* [21,22]. Moreover, both quantitative and qualitative disorders of the intestinal microbiota may occur during antibiotic therapy [23]. This particularly concerns patients receiving chronic high doses of antibiotics. In such cases, we usually deal with *Clostridium* superinfection [24]. Considering the above data, the model of our study seems to be the most reasonable.

## 2. The Aim of the Study

The objectives of our study are as follows: 

1. To evaluate the bacterial survival of several commercially available probiotic preparations as dietary supplements under conditions simulating the stomach environment (low pH) and the initial intestinal segment (bile). 

2. To evaluate the inhibitory capacity of four gastrointestinal pathogens by the probiotic formulations tested.

## 3. Material and Methods

### 3.1. Capsule Disintegration Study

The disintegration test of probiotic capsules was performed using disintegration apparatus: DisiTest 50, Dr. Schleuniger Pharmatron (Sotax), Swiss/USA. Experiments were carried out at a temperature of 37.0 °C ± 2 °C. The test was performed in a buffer of pH 2. The pH 2.0 buffer solution was prepared by dissolving 6.57 g of potassium chloride in water and adding 119.0 mL of 0.1 M hydrochloric acid, with water added to a volume of 1000 mL (Figure 1). The integrity of the capsules was observed at intervals of 5, 10, 15, 20, 25, 30, 40, 50, 60, 70, 80, 90, 100, 110, and 120 min.

### 3.2. Microbial Survival Study

Survival studies of probiotic microorganisms were conducted in three stages (Table 1). In this study, an experimental model was created to simulate “in vitro” acidic conditions in the stomach and in the presence of bile acid salts corresponding to their concentration in duodenal juice. Under these conditions (a solution of pH 2 equal to the acidity of the gastric juice of an adult), the first stage of the study determined the time after which disintegration of capsules containing different marketed probiotic products occurs. For this purpose, the capsules of market products were placed in a hydrochloric acid solution of pH 2. Additionally, the time after which the disintegration of the capsule wall and the release of its contents occurred was recorded.

In the second stage of the study, capsules with a known concentration of probiotic microorganisms were placed in hydrochloric acid pH 2 and incubated for 90 min. After this time, the number of live microorganisms present in the solution was determined. 

In the third stage of the study, the microorganisms contained in one capsule of market preparation were incubated for 180 min in 0.4% bile solution and the number of microorganisms after incubation was determined.

The study was conducted according to the methodology described in “Enumeration of probiotic microorganisms exposed to acid conditions” [25].

### 3.3. Examination of the Amount of LAB Bacteria before and after Exposure to Hydrochloric Acid

In the conducted experiments it was assumed that the tested capsules contained the concentration of probiotic microorganisms as declared by the manufacturer. The number of bacteria after incubation with hydrochloric acid and bile was measured using the plate method [26]. The results are presented as the arithmetic mean from three consecutive determinations.

### 3.4. Investigation of the Inhibition Zones of Marketed Probiotic against Pathological Bacteria

The four most common human gastrointestinal pathogens were used for this study.

*Eschericha coli*;*Shigella*;*Salmonella* spp.;*Clostridioides difficile* [27].

The gastrointestinal-pathogen-inhibitory abilities of microorganisms contained in five commercially available probiotic products were evaluated. To investigate the inhibitory effect of probiotics against pathological bacteria, experiments were performed by measuring the growth inhibition of these bacteria. Antagonism between microbiota was determined using the bar graph method according to Strus [28]. The quantitative results of inhibition of each probiotic product are presented as the arithmetic mean ± SD of three measurements obtained by inoculating pathological bacteria and determining the zone of inhibition (Table 1).

For in vitro growth inhibition studies with *C. difficile*, the pathogen was cultured under anaerobic conditions at 35–37 °C for 24–48 h on Schaedler agar (CM0437, Fisher Scientific GmbH, Schwerte, Germany). Suspensions, each containing 10^6^ CFU of each of the five products evaluated, were seeded onto MRS agar and incubated for 48 h in the presence of 5% CO_2_. Probiotic samples were transferred to Mueller–Hinton agar supplemented with 5% horse blood and 20 mg/L NAD (PP0972, E&O Laboratories Ltd., Bonnybridge, UK) and incubated under anaerobic conditions for 24 h. The zones of inhibition were measured as in the previous case.

## 4. Results

The longest time (60 min) after which the complete disintegration of the capsule, with the release of the probiotic into an environment imitating that of the stomach, occurred was observed for product A (Table 2, Figure 1). This product has an acid-resistant capsule with the highest resistance to hydrochloric acid. The time of the disintegration of regular capsules was from 10 min, products B and C, to 20 min for product D, and was 25 min for product E (Figure 2).

The greatest reduction in the number of live microorganisms after 90 min of incubation in hydrochloric acid at pH 2 was observed in the case of product C, in a capsule produced by traditional technology (Figure 3). This reduction was 3.12 on a logarithmic scale. The survival of microorganisms contained in product A, produced with acid-resistant capsule technology was the highest, and the reduction in the number of viable bacteria was the lowest, amounting to 1.08 on a logarithmic scale. The values of the reduction in the number of live microorganisms were comparable for all probiotic products produced with traditional technologies.

The reduction in the number of live microorganisms after 180 min exposure to 0.4% bile solution was similar in all tested probiotic products (Figure 4).

The reduction in live microorganisms after 180-minute exposure to 0.4% bile solution was similar in all tested probiotic products.

The largest zone of inhibition against pathological bacteria was found in cultures containing product A, produced by technology containing nine different bacterial strains (Figure 1 and Figure 5). The largest zones of inhibition were encountered with all four tested pathological bacteria (*Salmonella* spp. *E. coli, C. difficile*, and *Shigella* spp.). The smallest zones of inhibition were encountered in the case of probiotic products containing only one strain of probiotic bacteria. In this case, this was also observed for all four types of pathological bacteria. Intermediate sizes of inhibition zones occurred for the remaining probiotic products.

## 5. Discussion

It is well known that orally ingested food travels through the esophagus to the stomach. The enzymes and hydrochloric acid produced by the stomach are components of the gastric juice.

Under physiological conditions, food in the stomach is exposed to digestive juices and gastric motility for approximately 60 to 120 min [29,30]. These extreme environmental conditions are a requirement for proper digestion and provide a barrier to pathological microorganisms. The low pH of gastric juice disinfects the food and activates proenzymes [31]. In certain situations, this stage of digestion becomes unbeneficial for the human body. In cases where we want to intentionally introduce health-promoting substances, the gastric digestion process may reduce the expected effects of the administered preparations. An attempt to populate the gastrointestinal mucosa with microbiota administered in probiotics may serve as a typical example of these adverse interactions [9]. In order to populate further sections of the intestine, the process of “passage” through the stomach and duodenum should be taken into account and the preparations should be designed in such a way that the microorganisms survive and can reach the intestine alive.

There are 1012 bacteria in 1 g of colonic contents, making it the most colonized section of the gastrointestinal tract [32,33,34]. Bacteria colonizing the intestine create a complex ecosystem and having a great influence on the human body. The life processes of microorganisms produce metabolites that exert various effects on the host [35,36]. The extreme conditions in the stomach are an important barrier, the overcoming of which is a necessary condition for the proper development of bacterial colonies after the administration of probiotics from the outside [37]. The results of “in vitro” studies clearly suggest that the exposure of bacterial strains to concentrated acidic pH significantly affects their ability to survive in these adverse conditions. Based on the results of our study, it is clear that the most significant factor determining the high survival of microorganisms in conditions imitating the gastric environment is the type of capsule used during production and its susceptibility to concentrated hydrochloric acid. First of all, it is obvious that the time of disintegration and release of living microorganisms determine the later biological effect of a probiotic. A short time for disintegration of the probiotic capsule exposes the microorganisms contained within it due to the lack of a protective element, which in turn makes the microorganisms more susceptible to a low pH and affects their ability to survive these conditions.

Interestingly, even in the case of products with the highest content of live microorganisms, as declared by the producer, after their incubation for 90 min in hydrochloric acid, the amount of remaining live bacteria was lower than in preparations with a lower initial content. Obviously, this phenomenon was caused by the better protection of capsules made with a different technology. In our study, we showed that the exposure of microorganisms contained in probiotic preparations to bile acids contained in duodenal juice had less effect on their survival. 

These results are consistent with those obtained in the works of other authors [7]. Based on the results obtained, we can conclude that the most negative conditions for the survival of microorganisms take place in the gastric environment. The second important criterion for choosing the right probiotic preparation is undoubtedly the ability of the bacterial strains contained within to inhibit the development of pathological bacteria [17].

It should be noted that the formulations tested have different qualitative compositions of similar bacterial strains. This is important due to the fact that even bacteria from the same species may have different effects that are characteristic and specific to a given strain. The effectiveness of probiotics to inhibit the growth of pathological bacteria is shown by the size of the zone of inhibition measured on an agar plate. So far, the mechanisms responsible for these properties of probiotics have not been precisely determined in scientific studies. There is no doubt that we are dealing with an antagonistic effect of beneficial bacteria in relation to pathological bacteria. Currently, research is being conducted to explain this phenomenon. Probiotic microorganisms compete with pathological bacteria for space and nutrients. Bacteria in probiotics produce bacteriocins, substances that inhibit the colonization of potential pathogens [36,37]. This commensalism of probiotic microorganisms in the human body maintains a specific homeostasis, the disruption of which leads to gastrointestinal dysfunction. Based on the results obtained in our experiments, we can conclude that the most significant factor affecting the ability to inhibit the growth of pathological gastrointestinal bacteria is the diversity of bacterial strains contained in a given market product.

Enteropathogenic *E. coli*, EPEC, is a major cause of diarrhea in infants [38]. As long as there is no evidence of systemic infection, antibiotic therapy is rarely indicated and should be delayed until culture results are available. For this reason, and because of the emerging antibiotic resistance of *E. coli* [39], probiotics are being considered as additional treatment options for *E. coli* infections [40]. In previous in vitro pathogen growth inhibition experiments, no clear inhibition of *E. coli* growth by *S. Boulardii* yeast has been reported [41]. In contrast, the *in vitro* growth inhibition of *E. coli* has been described for many single-strain bacterial probiotics, among them *L. rhamnosus* GG [42] and *L. reuteri* DSM 17938, and multi-strain probiotics [43,44]. Intestinal microbiota disorders also affect the oral cavity, which is a challenge in dentistry [45]. The situation is similar for other intestinal pathogens (*Shigella, Salmonella* spp, and *C. difficile*), and probiotics may also be beneficial in these cases. Questions related to bacterial flora disorders dependent on antibiotic therapy have been asked for a very long time. In recent years, studies have found that dysbiosis can also occur in cases of therapy with drugs such as metformin, nonsteroidal anti-inflammatory drugs, or proton pump inhibitors [46,47,48]. It is worth emphasizing that the use of these drugs may be associated with the development of intestinal pathogens such as *E. coli, Shigella, Salmonella* spp, and *C. difficile*. This is important because these drugs are used in the population in very large amounts.

These facts show the importance of choosing the right probiotic formulation to obtain the best clinical effect. Further studies of both single-bacterial-strain probiotics and combinations of probiotics are needed to obtain the optimal formulation in a given clinical situation.

## 6. Conclusions

The survival of microorganisms contained in market products of probiotics depends mainly on the type of capsule used in the production and the time of its disintegration. The fast disintegration of the capsule and the low survival rate of the microorganisms contained within it contribute to the weaker colonization of the intestine and the lower effectiveness of their action. It can be clearly stated that the greater the diversity of bacterial strains in a given probiotic preparation, the greater the ability of this population to inhibit pathological bacteria under “in vitro” conditions. This relates to all four gastrointestinal pathological bacteria we tested (*Salmonella* spp., *Escherichia coli, Clostridioides difficile*, and *Shigella* spp.) Of the market products tested, Multilac^®^ showed the best survival rate and the best antimicrobial properties.

## Figures and Tables

**Figure 1 nutrients-14-03323-f001:**
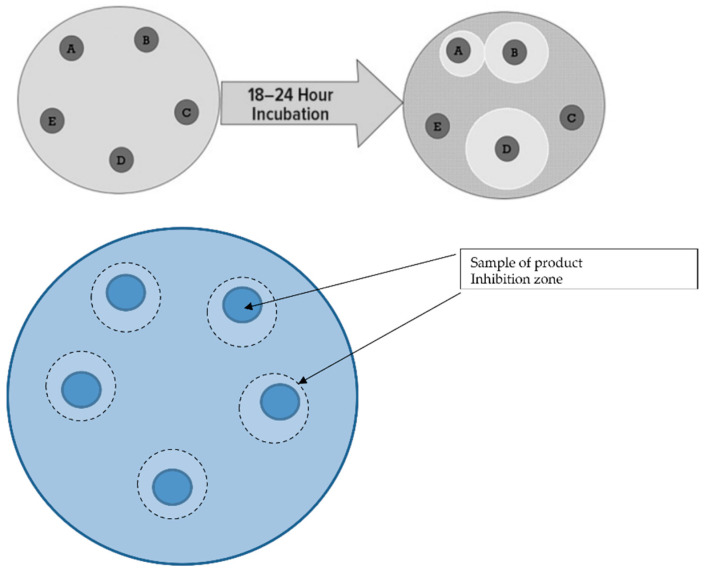
Schematic of the culture medium for pathological bacteria and probiotic market products. Explanation: product A (*Lactobacillus helveticus, Lactococcus lactis, Bifidobacterium longum, Bifidobacterium breve, Lactobacillus rhamnosus, Strepto-coccus thermophilus, Bifidobacterium bifidum, Lactobacillus casei, Lactobacillus plantarum*), product B (*Lactobacillus plantarum*), product C (4 *Lactobacillis,* 2 *Bifidobacterium*, *Lactococcus lactis*), product D (*Lactobacillus rhamnosus*), product E (4 *Lactobacillis*3 *Bifidobacterium, Streptococcus thermophilus*).

**Figure 2 nutrients-14-03323-f002:**
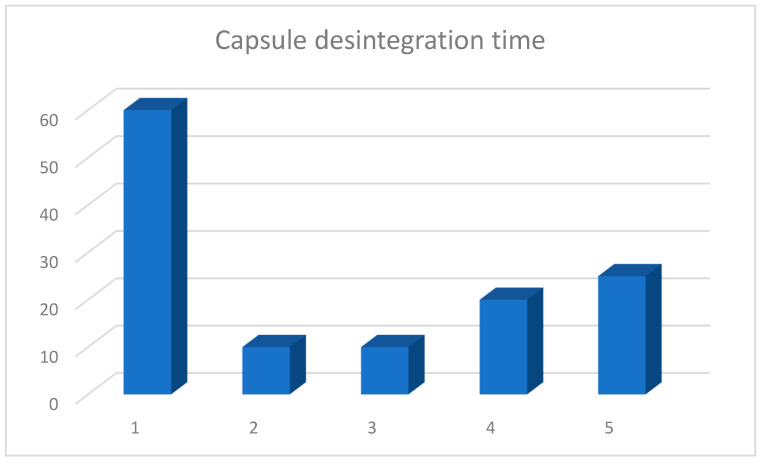
Average disintegration time of capsules containing market products of probiotics. Arithmetic means of three independent experiments.

**Figure 3 nutrients-14-03323-f003:**
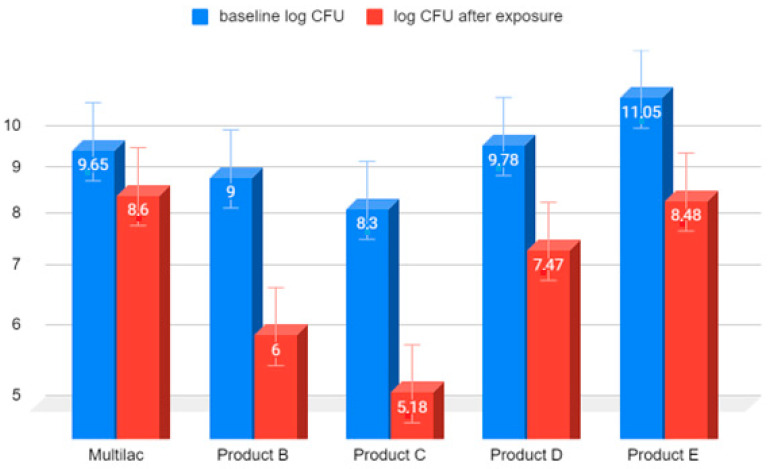
Comparison on a logarithmic scale of the decrease in the number of live LAB bacteria in market preparations of probiotics after 90 min of exposure to HCl solution at pH 2. Arithmetic means of three independent experiments.

**Figure 4 nutrients-14-03323-f004:**
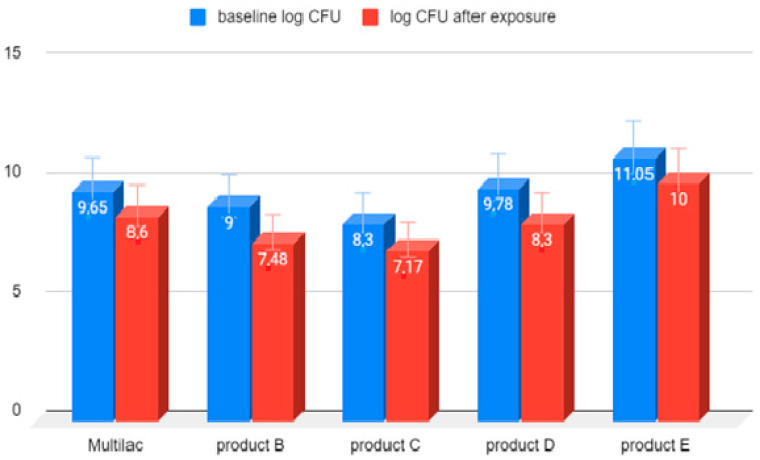
LAB bacterial counts before and after 180 min of exposure to 0.4% bile solution. Arithmetic means of three independent experiments.

**Figure 5 nutrients-14-03323-f005:**
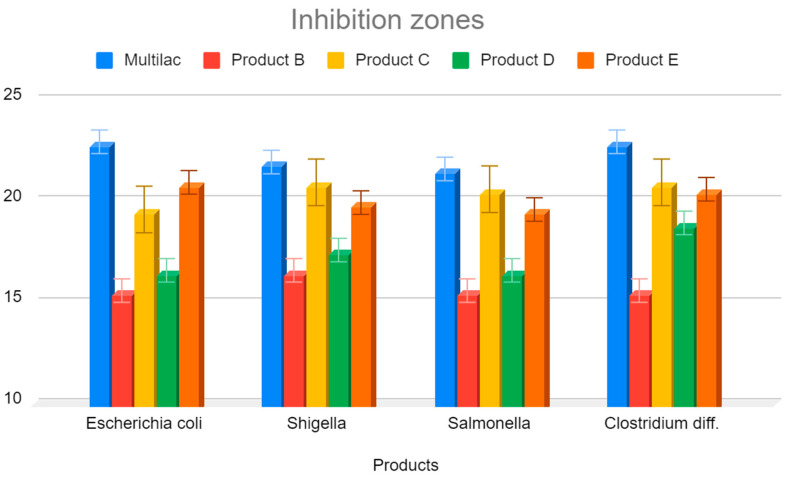
Zones of inhibition of tested probiotics against pathological gastrointestinal bacteria. Arithmetic mean of 3 independent experiments.

**Table 1 nutrients-14-03323-t001:** Probiotic products used in the study.

Product	Bacterial Composition of the Product	Capsule Type
Product A (Multilac^®^) 4.5 × 10^9^ bacteria in the capsule	*Lactococcus lactis Ll-23 13* × *10^8^ CFU, Lactobacillus plantarum LP-115 2.5* × *10^8^ CFU, Lactobacillus rhamnosus BI-FOLAC™ GG 1.5* × *10^8^ CFU, Streptococcus thermophilus ST-21 1.1* × *10^8^ CFU, Bifidobacterium breve BB-03 1* × *10^8^ CFU, Lactobacillus casei Lc-11 0.4* × *10^8^ CFU, Bifidobacterium bifidum Bb-02 1* × *10^8^ CFU, Bifidobacterium animalis ssp. lacti BIFOLAC™ 12 21.5* × *10^8^ CFU, Lactobacillus acidophilus LA-14 2* × *10^8^ CFU*	Gastro-resistant capsules
Product B 1 × 10^9^ bacteria in the capsule	*Lactobacillus plantarum*	Regular capsules
Product C 2 × 10^8^ bacteria in the capsule	4 *Lactobacillis* 2 *Bifidobacterium* *Lactococcus lactis*	Regular capsules
Product D 6 × 10^9^ bacteria in the capsule	*Lactobacillus rhamnosus*	Regular capsules
Product E 1.12 × 10^11^ bacteria in the capsule	4 *Lactobacillis* 3 *BifidobacteriumStreptococcus thermophilus*	Regular capsules

**Table 2 nutrients-14-03323-t002:** Capsule disintegration time in hydrochloric acid at pH 2 in minutes.

Multilac^®^	Product B	Product C	Product D	Product E
60	10	10	20	25

## Data Availability

Not applicable.

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
