# Peer review of "Factors Determining Effective Probiotic Activity: Evaluation of Survival and Antibacterial Activity of Selected Probiotic Products Using an “In Vitro” Study"

_nutrients, 2022, doi:10.3390/nu14163323_

Round 1

Reviewer 1 Report

the manuscript deals with an essential and interesting topic. the microbiome and its maintenance plays an essential role. Therefore, the research projects and the results are very relevant. I find it irritating that only one product was mentioned in the article in comparison, which was summarized as the best in the last sentence. how could you ensure that the results were not influenced by the sponsorship. did the researchers know the contents of the samples? Are there any comparative papers with other results? Please describe this points 

Author Response

Thank you very much for completing the review of our publication. I would like to clarify that the laboratory staff performing the experiments received probiotic products in unlabeled capsules. Consequently, they did not know which capsules contained which probiotics. This guaranteed that the study was performed objectively. We are currently testing further market products of probiotics.

Reviewer 2 Report

Dear Authors,

Your submitted manuscript is very necessary and practise. Keep it up!

I have only a few more or less technical comments. see below:

Line 47 - [7], [8] change to [7, 8]

„Lactobacillus spp., Bifidobacterium spp., Clostridium spp., 62 Bacteroides spp“- the latin names must be in italic

Line 78 – 79 – This sentence is not logically arranged. It is not related in meaning to the previous sentence.

Line 86 – change „Salmonell“ to „Salmonella

Line 86 – all latin names of strain must be in italic the same is right for Table 1 and all text

Please unify the citation of the used literature in the references.

Author Response

Thank you very much for your careful review of our work. In the final version of the article we will take into account all the objections and errors.

Reviewer 3 Report

The manuscript is well written although I have a series of doubts that I need to clarify.  There are minor points, typos or curiosities, that should be revised, explained and, if necessary, corrected. 

Line 47, 53 and 284. According to the Vancouver style standard the bibliography is incorrectly written.

The entire text must be checked when words are separated in the subsequent line. Example line 48, line 59, etc.

In the introduction, gut microflora, intestinal microflora, bacterial flora, etc. Are used as synonyms. A more appropriate term such as gut microbiota should be used.

There are updates to the term probiotic that should be included in the bibliography of the paper. Hill et al. 2014. The International Scientific Association for Probiotics and Prebiotics consensus statement on the scope and appropriate use of the term probiotic.

There is a current reclassification of the generic term Lactobacilli that reflects the phylogenetic position of microorganisms, and groups Lactobacilli into robust clades with shared ecological and metabolic properties. Zheng et al. 2020. Please, check which genres have been renamed. Also check the Clostridium difficile change. The Lancet Infectious Diseases Volume 19, Issue 5, May 2019, Page 449. Revise italic writing of the probiotic species named.

Figures and tables are not explained in the text. Figure 1 is disproportionately sized in the text of the paper.

Product A (multilac) has 4.5 x 109 CFU. Products B, C, D and E have what number of CFU?

I don't understand the line 143 “...bacterial strains were presented.....” Are products A, B, C, D and E probiotic strains (I don't know which ones) or are they probiotic species? Please, could you clarify the doubt?

According to figure 1 the largest zone of inhibition is for product D (a single probiotic species). However, line 204 says the opposite. Would you please explain to me which is correct?

Line 192. What does the author mean by "traditional technologies"?

Could you please explain these concepts more clearly in your paper?

Resume

It is true that the viability of probiotics depends on the type of capsule, speed of disintegration, number of probiotic strains, etc. However, the authors do not consider other parameters such as water activity, particle size, microcapsule shape, encapsulation with different inorganic materials, etc.

I think that these parameters should be mentioned in the discussion and the conclusion would be more accurate. On the other hand, without taking all these parameters into account, it is adventurous to say that multilac (is it a trade name?) has the best effectiveness of the 5 products.

Author Response

Thank you very much for your careful review of our work. With all the comments we completely agree. In the final version of the article we will try to take into account all the objections and suggestions and correct the errors. Figure 1 is only a diagram showing the conduct of the experiment but not the results we obtained.

There is a mistake in line 143. It should be :

The quantitative results of inhibition of each probiotic product are presented as the arithmetic mean ± SD of three measurements obtained by inoculating pathological bacteria and determining the zone of inhibition (Table 1).

This has been corrected in the final version of the publication.

"traditional technologies" - for this phrase, we meant capsules that dissolve in the stomach.

Multilac is trade name.